# Approaches to Identify and Characterise the Post-Transcriptional Roles of lncRNAs in Cancer

**DOI:** 10.3390/ncrna7010019

**Published:** 2021-03-09

**Authors:** Jean-Michel Carter, Daniel Aron Ang, Nicholas Sim, Andrea Budiman, Yinghui Li

**Affiliations:** 1School of Biological Sciences (SBS), Nanyang Technological University (NTU), 60 Nanyang Drive, Singapore 637551, Singapore; DANI0045@e.ntu.edu.sg (D.A.A.); YANGUNIC001@e.ntu.edu.sg (N.S.); andrea.budiman@ntu.edu.sg (A.B.); 2Institute of Molecular and Cell Biology (IMCB), A*STAR, Singapore 138673, Singapore

**Keywords:** lncRNA, cancer, post-transcription, RNA-binding, ribonucleoprotein, RNAi, interactome, prediction, database, CLIP

## Abstract

It is becoming increasingly evident that the non-coding genome and transcriptome exert great influence over their coding counterparts through complex molecular interactions. Among non-coding RNAs (ncRNA), long non-coding RNAs (lncRNAs) in particular present increased potential to participate in dysregulation of post-transcriptional processes through both RNA and protein interactions. Since such processes can play key roles in contributing to cancer progression, it is desirable to continue expanding the search for lncRNAs impacting cancer through post-transcriptional mechanisms. The sheer diversity of mechanisms requires diverse resources and methods that have been developed and refined over the past decade. We provide an overview of computational resources as well as proven low-to-high throughput techniques to enable identification and characterisation of lncRNAs in their complex interactive contexts. As more cancer research strategies evolve to explore the non-coding genome and transcriptome, we anticipate this will provide a valuable primer and perspective of how these technologies have matured and will continue to evolve to assist researchers in elucidating post-transcriptional roles of lncRNAs in cancer.

## 1. Introduction

Transcription is at the forefront of the conversion of stable genomic information into reactive biochemical agents that form and modulate dynamic biological systems. This fundamental process relentlessly transcribes at least 62% of the human genome, resulting in a variety of non-coding RNA (ncRNAs) species that outnumbers the selection of more stable RNAs concerned with translation that accumulate in the cell such as ribosomal RNA (rRNA), transfer RNA (tRNA) and messenger RNA (mRNA) [1,2]. Far from being redundant transcriptional byproducts, ncRNAs can also act as pleiotropic reactive biochemical agents interacting with both RNAs and proteins and are first to propagate any genome level information changes to the biological network state that shapes cellular behaviour [3,4].

Diseased states such as cancer arise from cumulative corruption of the genomic source code, resulting in the opportunistic dysregulation of the conserved transcriptional [5], post-transcriptional [6], translational [7] and post-translational processes [8] that ultimately allow them to escape systemic control. Among the hundreds of thousands of genetic abnormalities that may occur, tremendous progress has been made in understanding how specific key “driver” mutations affect important protein coding genes (oncogenes and tumour suppressors) and influence the aforementioned processes to provide a selective advantage to cancerous cells to develop in a given tissue or microenvironment [9]. However, many mutations found in cancer also accumulate in the non-coding genome [10,11].

The large non-coding transcriptional contributions such as small non-coding RNA (sncRNA), enhancer RNA (eRNA) and long non-coding RNAs (lncRNA) have come under increased interest for cancer research. They feature prominently among the rapidly increasing list of non-coding regulatory elements vulnerable to mutations (promoters, enhancers) and contributing to dysregulating the critical processes aforementioned [12,13]. Although some of these ncRNAs, such as microRNAs (miRNAs, a subset of sncRNAs) have garnered plenty of research momentum [14], others such as lncRNAs and circular RNAs (circRNAs) are still burgeoning especially in the context of cancer biology [15].

Long non-coding RNAs (≥200 bp; lncRNAs) encompass the largest and perhaps most intriguing category of ncRNAs in cancer currently known, as they exhibit highly dynamic and tissue specific expression patterns [16], a trait shared with most oncogenes/tumour suppressors [17]. The majority of these transcripts are localised in the nucleus and transcribed by RNA Polymerase II (RNA Pol II). Long ncRNAs share similar characteristics to messenger RNAs (mRNAs), such as having a 5′-cap and 3′ poly-A tail. Alternatively, circular forms may assemble through non-coding splicing of exons and introns (circRNA), bolstering their resistance to degradation further [18,19]. Classification of the diversity of ncRNAs, especially lncRNAs, is a work in progress. GENCODE has classified lncRNA annotation on the basis of their genomic context, however this offers little indication of their diverse functional potential aside from identifying possible antisense lncRNAs [20,21,22]. Interestingly, the length of an RNA has been found to correlate with their propensity to interact with other biochemical molecules, such as proteins [23]. Hence, the “longer” spectrum of ncRNAs may well be more prone to assume varied roles in regulatory processes through cross-molecular interactions. Indeed, many have already been found to play a pivotal role in those exploited by cancers, such as development and differentiation of cells [24,25]. In fact, since the discovery of MALAT1 (Metastasis-associated lung adenocarcinoma transcript 1) in the early 2000s [26], dysregulation of more than a dozen other lncRNAs, such as *H19* [27], *XIST* [28,29], and *HOTAIR* [30,31] have steadily been found to be associated with cancer progression and drug resistance [32,33,34]. There is also evidence of circRNAs playing a role in cancer progression [35] and chemotherapy resistance along with biomarker potential [36]. With approximately 16,000 lncRNA genes (28,000 transcripts) identified in Gencode 27, this may well represent a potential “goldmine” of hidden tumour suppressor/oncogenic targets [37].

A major challenge remains in uncovering these lncRNAs and revealing their functional roles in cancer-exploited regulatory processes. This is principally due to the novelty of the field, which is further complicated by their multifunctional interactive potential. At least four major mechanisms have been suggested to mediate their effects: (1) Act as signals to regulate transcription, (2) as decoys recruiting binding partners away from their other targets, (3) as guides directing the targeting of a ribonucleoprotein complex, for example; (4) as scaffolds bringing together multiple biomolecules together [38,39]. Such mechanisms are directly susceptible to propagating abnormalities in lncRNA expression or sequence typical of cancer to the post-transcriptional regulatory networks [40]. Mechanisms 2 and 3 are known to influence one of the most prominent post-transcriptional regulatory pathways: miRNA mediated RNA interference (RNAi) [41]. Effectively, lncRNAs may act as either target or ordnance for the RNA-induced silencing complex (RISC) that perturb stability of various RNAs including lncRNAs themselves [42]. Additionally, lncRNAs may interact with RNAs directly as antisense lncRNAs, such as KRT7-AS, which promotes gastric cancer progression [43]. In summary, aberrant lncRNAs may exert substantial influence over post-transcriptional dysregulation in cancer through RISC dependent and independent mechanisms mediated via their interactions with RNA-binding proteins (RBPs) or RNAs [44]. Identifying and characterising these interactive mechanisms utilising appropriate approaches is therefore critical to overcoming the aforementioned challenge in elucidating their roles in major regulatory processes such as post-transcription.

In this review, we will provide a survey of the most useful tools and techniques developed to help place lncRNAs on the post-transcriptional interactome map and reveal their roots to cancer. In the first part, we will provide an overview of such resources valued for primary identification and characterisation of lncRNAs, especially those capable of highlighting cancer relevant contextualisation. In the second part, we will cover how more advanced resources have been developed to help characterise how lncRNAs may interact with RNAs and proteins (Figure 1). Ultimately, we hope this will serve as a useful primer for new cancer research strategies interested in identifying and validating further lncRNAs as oncogenic/tumour suppressor-like players by mechanistically uncovering their post-transcriptional roles.

## 2. Identification and Primary Characterisation

### 2.1. Predictions, Identification from High-Throughput Data and Databases

Most cancer research strategies begin by identifying potential candidate genes/loci involved in the dysregulatory state under study. Screening for such candidates usually involves the intersection of high-throughput screening experiments, especially next-generation (NGS) or third-generation sequencing [45]. A number of RNA-sequencing techniques can directly provide valuable information on identity, expression and/or stability of RNAs including lncRNAs. These include capturing a sample of the total transcriptome via standard or single cell RNA-sequencing (RNA-seq/scRNA-seq); capturing nascent transcription using tagged nucleosides or analogs through Global or Precision Run-On sequencing (GRO-seq/PRO-seq; [46,47]) as well as Bru-seq/Bru-Chase-seq [48]; capturing full-length transcripts through Cap-analysis gene expression sequencing (CAGE-seq) and nano-cap analysis of gene expression (nanoCAGE+CAGEscan; [49]) or Oxford Nanopore native RNA sequencing [50]. Provided such sequencing datasets do not undergo selective library preparations these can allow the identification of lncRNAs directly by sequence. Other scenarios, such as Chromatin immunoprecipitation (ChIP) followed by high-throughput DNA sequencing or Assay for Transposase-Accessible Chromatin using sequencing (ChIP-seq or ATAC-seq) may lead towards selecting a putative non-coding region of the cancer genome [51,52]. Depending on the model system, experimental designs and resources, it will be worth considering whether performing additional sequencing experiments is really necessary to fulfil research objectives considering the large amounts of publicly available second-generation data already available.

Regardless of which experimental strategies have been implemented, orthogonal lines of evidence will always fall back on a specific locus or multiple loci of the non-coding genome. From this point, it is possible to infer the identity of potential linear lncRNAs by cross-referencing the loci coordinates with several large databases dedicated to cataloguing lncRNAs or predicting the coding potential of the region. If the coding potential has not already been evaluated, numerous machine-learning tools are available to perform this computation de novo. CPAT, FEELnc and PLEK may be particularly suitable for working with human cancer datasets as extensively evaluated alongside numerous other solutions in [53]. Further comparative reviews of the features of such tools can also be found in [54].

In the case of circular RNAs, most achieve their circular conformation via “backsplicing”. This refers to the covalent linkages between a downstream 3’ and an upstream 5’splice sites, which results in a reversal of exon sequences relative to the annotated transcript [55]. This unique mechanism can be exploited for their identification and therefore a number of tools have been developed to perform this on RNA-seq datasets, which are extensively evaluated and reviewed in [56,57]. CIRI and KNIFE are among some of the tools that showed robust performance even among background noise [58,59]. CIRI2 has also recently been released offering significant performance improvements over CIRI [60].

In the vast majority of cases—especially for cancer studies based on the human genome—a wealth of sequencing data, pre-generated predictions and annotations relevant for targeted loci are available from a number of public and restricted access databases. Aside from familiar initiatives such as RefSeq, Ensembl and FANTOM [61], many more specialised resources dedicated to allocating lncRNA identity and valuable annotations have emerged—some of which are tailored to cancer research, such as Lnc2Cancer or CSCD. Beyond simply determining whether the loci of interest is transcribed as a lncRNA, many of the resources presented offer insights into transcript localisation, expression as well as gene conservation, mutation and links to diseases, such as cancer. Much of the information is integrated from other public databases and projects such as ClinVar [62], COSMIC (Catalogue of Somatic Mutations in Cancer) [63], TCGA (The Cancer Genome Atlas) [64], 1000 Genomes Project (IGSR) [65], (G)ENCODE (Encyclopedia Of DNA Elements) [66,67], GEO (Gene Expression Omnibus) [68], dbSNP [69], UniProt [70], HPA (Human Protein Atlas) [71], GTEx (Genotype-Tissue Expression) [72], HBM2 (Human Body Map 2.0 GEO Dataset GSE30611), FANTOM (Functional Annotation of the Mammalian Genome) [73], CCLE (Cancer Cell Line Encyclopedia) [74], Disease Ontology [75], GO (Gene Ontology) [76], MeSH (Medical Subject Headings) [77] and TARGET (Therapeutically Applicable Research to Generate Effective Treatments) [78]. Additionally, many circRNA specialised databases integrate information and predictions from miRNA centric databases, since circRNAs often act miRNA sponges. These include Starbase [79], TargetScan [80], doRiNA [81], miRcode [82], miRTarBase [83], HMDD [84], OncomiRDB [85], dbDEMC [86] and miRecords [87]. Although most resources will require usage of the web interface, some offer more advanced programmatic access such as NONCODE [88]. All lncRNA and circRNA resources are summarized in Table 1 and Table 2 respectively.

While these databases offer an increasingly large and reliable set of annotations, they remain putative until validated in the specific cancer model system under study using the experimental approaches covered below.

### 2.2. Experimental Approaches: Validation of Expression, Localisation & Structure

Before any advanced experiments take place, it is usually preferable to validate basic characteristics of the target lncRNAs of interest. We will therefore briefly highlight some key primary techniques valued for lncRNA characterisation as well as potential limitations in their application.

**Northern blotting** has long been applied for analysing expression of specific RNAs, enabling relative quantification, determination of sizes and providing an assessment of the RNA quality [124,125]. Modern protocols allow reduced chemical usage and good specificity and this core method still remains a vital tool for primary characterisation of lncRNAs [126]. In addition, this technique is still one of the most direct methods for demonstrating the circular configuration of circRNAs. Furthermore, the method is often integrated with more advanced procedures to investigate ribonucleoprotein complexes [127].

Providing the target RNA may be reverse transcribed, **RT-qPCR** (reverse transcriptase polymerase chain reaction) may offer a more convenient high sensitivity assay. The exponential nature of qPCR, however, requires careful consideration of confounding factors such as genomic DNA contamination and appropriate selection of reference genes [128]. The latter can still be quite problematic when the system of study features aberrant expression of multiple genes including house-keeping genes commonly chosen as references. The recent availability of large pan-cancer datasets may be able to help overcome this problem [129].

Another valuable technique, especially for characterising unknown isoforms of lncRNAs suspected of undergoing splicing is rapid amplification of cDNA ends (RACE), which produces full length sequences of RNA transcripts. RACE utilises reverse transcription with a 5′ or 3′ primer of a known sequence of the RNA of interest to produce a cDNA copy, this is then followed up with PCR amplification. The product can then be coupled with high-throughput screening in a technique called RACE-Seq to characterize the RACE fragments [130,131].

**RNA fluorescence in situ hybridization** (RNA-FISH) is the reference technique for visually detecting and determining the distribution of any type of RNA in cellular compartments as well as cells that express the RNA of interest. This technique uses fluorescently labelled probes specific for the target RNA [132,133]. While this technique has been crucial in uncovering the mechanism of several lncRNAs [134,135], the high amounts of repetitive elements in lncRNAs increases the challenge of detecting a legitimate lncRNA signal. This may result in the probe binding to high-abundance, off-target RNAs instead of the intended lncRNA. Furthermore, lncRNA signals in the nucleus appear as “bright blobs”, which can be difficult to differentiate from non-specific background signals [136].

**SHAPE** (Selective 2′-hydroxyl acylation analysed by primer extension) involves the use of reagents, such as N-methylisatoic anhydride (NMIA) and 1-methyl-7-nitroisatoic anhydride (1M7) that react with the 2′-hydroxyl group of the RNA backbone, forming ribose 2′-O-adducts [137]. Adduct formation is dependent on nucleotide flexibility and is quantified at nucleotide resolution by performing RT and comparing the product against a control [138]. This can be further coupled with mutational profiling (MaP), which accounts for the occasional incorporation of noncomplementary nucleotides or deletions caused by reverse transcriptase enzymes, to generate SHAPE profiles where mutations are counted and facilitate the identification of RNA secondary structure formation at nucleotide resolution [139,140]. With valuable evidence supporting the expression, localisation and possible structure of the target lncRNAs, the next step is to estimate and conclusively identify what biomolecules may be interacting with it.

## 3. Secondary Characterisation: Predicting and Detecting Interactions

### 3.1. Predictions and Databases

As mentioned in the introduction, there are four types of molecular mechanisms suggested to mediate lncRNA effects through versatile interactions with DNA, RNA and protein molecules [141]. All three types of interactions have been studied and modelled. However, the interactions with the most direct effects on post-transcription are expected to involve only RNA and protein. For more information on predicting lncRNA:DNA interaction potential, we invite the reader to consult recent developments in this relatively new field [142,143,144].

In-silico prediction of RNA-RNA and RNA-protein interactions are active areas of research recently boosted by machine learning techniques that have grown in strength and numbers over the past decade, feeding on the wealth of accumulating experimental data [145,146]. A significant amount of time and resources have been invested into developing specialised databases and algorithms that predict potential interacting partners of particular lncRNA candidates, some of which have required supercomputer scale processing [147,148]. Exploring how some of these resources can help guide or supplement experimental approaches should therefore form a valuable addition to the secondary characterisation strategy for lncRNA candidates.

RNA:RNA interaction prediction stands to be the most well investigated in large part due to the strong overlap with the small RNA/miRNA field. For instance, lncRNAs acting as ceRNAs can be predicted through their interactions with miRNAs for which a multitude of databases and tools already exist to predict their general propensity to bind certain RNA sequences (Table 3); also see [149]. However, some lncRNA:RNA specific prediction tools have been developed too. Recent evaluation of a dozen such tools has shown that the real-life performance is still fairly average [53]. Tools such as ASSA or RIblast [150,151] that incorporate other sequence information, such as length and GC content and provide useful statistical outputs may be most relevant for real human datasets but cannot be solely relied on for confident inference.

RNA:Protein prediction requires a different approach. Notably, network and correlation based predictions [152] have gained popularity owing to the large increase in available expression data, allowing indirect inference of lncRNA:Protein relationships. RNA:RNA prediction can also benefit from this approach and should be used to complement predictions [153,154]. As such, most resources presented are databases integrating multiple sources of evidence from orthogonal experiments in repositories such as GEO [68], ArrayExpress [155], ENCODE [67], TCGA [64] and the SRA (Sequence Read Archive) [156] as well as pre-calculated prediction and annotation databases such as as LncRNADisease [94], MNDR (Mammal ncRNA-Disease Repository) [97], eDGAR (Database of Disease-Gene Associations with annotated Relationships among genes) [157], circRNADisease [114], RAIN (RNA-protein Association and Interaction Networks) [158], RAID (RNA Association Interaction Database) [159], NPInter [160] and RISE (RNA Interactome from Sequencing Experiments) [161] (Table 3). For a comprehensive review of tools and databases dedicated to miRNA specific predictions please refer to [149,162].

**Table 3 ncrna-07-00019-t003:** Databases integrating computational and experimental sources for predicting lncRNA interactions. Databases are listed detailing the types of interactions they cover. A link to the hosting website is provided followed by the latest known version as well as the most recent publication describing the database. Most databases rely on a primary experimental (EXP) or computational (CPU) source, which are briefly explained. Any additional sources are also summarised.

Database	Link	Interaction Type	Primary Source	Additional Sources
NPInter v4 (2019) [160]	http://bigdata.ibp.ac.cn/npinter4 (accessed on 8 March 2021)	miRNA-RNA; ncRNA-DNA; ncRNA-Protein; circRNA	EXP: Re-processing and integration of experimental data (**GEO; ENCODE; RISE**)	CPU: miRNA binding (**miRanda, TargetScan**); Disease association (**LncRNADisease**, **MNDR**, **eDGAR** and **circRNADisease**)EXP: Literature mining
lncRRIsearch (2019) [163]	http://rtools.cbrc.jp/LncRRIsearch/ (accessed on 8 March 2021)	lncRNA-mRNA	CPU: RIBlast	EXP: Tissue expression
DIANA-LncBase v3 (2020) [164]	https://diana.e-ce.uth.gr/lncbasev3 (accessed on 8 March 2021)	miRNA-lncRNA	EXP: Re-processing and integration of experimental data (miRNA, AGO2-CLIP-Seq and CLIP-Seq)	CPU: Correlation with lncRNA expression
SPONGEdb v1 (2021) [165]	https://exbio.wzw.tum.de/sponge/home (accessed on 8 March 2021)	miRNA-lncRNA	CPU: **DIANA-LncBase**, **TargetScan**, **miRcode**, **miRTarBase**	EXP: TCGA expression
LnCeVar v1 (2020) [166]	http://www.bio-bigdata.net/LnCeVar/ (accessed on 8 March 2021)	miRNA-lncRNA	EXP: SNP and mutation data from **TCGA**, **COSMIC**, **1000 Genomes Project**	CPU: Integration from **miRanda, mirBase, miRTarBase, TargetScan**
miRSponge v1 (2015) [167]	http://bio-bigdata.hrbmu.edu.cn/miRSponge/ (accessed on 8 March 2021)	miRNA-lncRNA miRNA-circRNA	EXP: Manual curation from literature	CPU: Integration from **TarBase, miRTarBase, miRanda, miRecord**
starBase/ENCORI v2 (2014/2021) [79]	http://starbase.sysu.edu.cn/ (accessed on 8 March 2021)	miRNA-ncRNA; RBP-RNA;RNA-RNA	EXP: Re-processing and integration of experimental data (CLIP-Seq & variations)	CPU: Correlation of RBP somatic mutation with diseases EXP: Pan-Cancer networks from expression profiles (**TCGA**)
RAID v3/RNAInter (2020) [168]	http://www.rna-society.org/raid/ (accessed on 8 March 2021)	RNA-Protein; RNA-RNA; RNA-Histone; RNA-Drug	EXP/CPU: Integration of literature sources and 35 databases.	EXP: Methylation, localisation and editing data from other databases.
RISE (2018) [161]	http://rise.life.tsinghua.edu.cn/index.html (accessed on 8 March 2021)	RNA-RNA	EXP: Integration from sequencing based studies	CPU: Integration with several other databases (**RAIN**, **RAID**, **NPInter**)
LncRNA2Target v2 (2019) [169]	http://123.59.132.21/lncrna2target/ (accessed on 8 March 2021)	lncRNA-RNA	EXP: Manual extraction of interaction associations from literature	EXP: Re-processing of lncRNA perturbation RNA-Seq datasets
LncExpDB (2020) [170]	https://bigd.big.ac.cn/lncexpdb/interactions (accessed on 8 March 2021)	lncRNA-mRNA	CPU: Co-expression network analysis and prediction	EXP: Expression extracted from public repositories (**GEO**, **SRA** and **ArrayExpress**)
LncACTdb v2 (2019) [171]	http://www.bio-bigdata.net/LncACTdb (accessed on 8 March 2021)	miRNA-lncRNA-mRNAmiRNA-circRNA	EXP: Manual curation from literature	CPU: Predictions from networks and integration with Pan-Cancer data (**TCGA**)

These tools and databases should help prioritise the types of hypotheses and experiments planned for experimental validation in the cancer model system of choice utilising several of the numerous techniques covered in the rest of this article.

### 3.2. Sequencing Compatible Approaches

With the advent of next-generation sequencing, multiple high-throughput techniques have been developed which allow for increased screening capabilities to discover novel lncRNA interactions in cancer. One of the primary mechanisms through which lncRNAs have been documented to dysregulate cancer post-transcription is through their participation in ribonucleoprotein (RNP) complexes [44,172]. Considering the importance of RBPs, we will first introduce the RIP and CLIP methods, which have been more recently adapted to RNA-Seq. We will then focus on how related methods have been specifically tailored for capturing RNA interactions involving the RNA-induced silencing complex (RISC). Finally, methods for exploring RISC independent RNA interactions will be presented.

#### 3.2.1. Ribonucleoprotein Complex Interaction Detection

LncRNAs often interact with RNA-binding proteins (RBPs), such as the interaction between *HOTAIR* and *EZH2* [173]. Several knock-on effects can result, such as competing with other mRNAs/lncRNAs for RBP binding and/or increasing the ceRNA potential of the lncRNA. In the *MACC1-AS1* to *PTBP1* interaction, such effects have significant consequences on breast cancer tumorigenesis [174]. RNA immunoprecipitation (**RIP**) is one of the first techniques employed to identify such RNAs bound to specific RBPs. RIP involves cell or tissue lysis, followed by immunoprecipitation of native RNA-protein complexes with a specific antibody against the target protein. As these complexes are not stabilized by covalent crosslinking, extra precaution must be taken during washing to remove nonspecific RNA while maintaining the RNA–protein interactions. This limitation makes detection of RNAs with low binding affinity to the protein of interest difficult. In addition, unstably bound RBPs may dissociate from their RNA targets and re-associate with other RNAs under harsh conditions [175,176].

Nevertheless, the RIP techniques have been successfully used over the years revealing relevant interactions in the context of cancer. For example, RIP was used by Tripathi et al. to investigate the interaction between lncRNA metastasis-associated lung adenocarcinoma transcript 1 (*MALAT1*) and the serine/arginine (SR) splicing factors [177]. *MALAT1* is known to be overexpressed in breast, pancreas, lung, colon and prostate carcinomas [178], in addition, it is associated with metastasis and poor survival in non-small cell lung cancer patients [26]. SR splicing factors can influence the alternative splicing (AS) events of many pre-mRNAs in a concentration and phosphorylation-dependent manner, but its cellular mechanism was unknown [179,180,181]. In the study, *MALAT1* was found to interact with SRSF1 and regulate cellular levels of its phosphorylated forms, which modulated AS events downstream. Further exploration of *MALAT1* interactions via **RIP-Seq** by Wang et al. was similarly fruitful. Their study revealed that *MALAT1* also binds to *EZH2* [182]. *EZH2* is overexpressed in endometrium, prostate and breast cancers [183]. In prostate cancer patients, *EZH2* is associated with increased cell proliferation, invasiveness and metastasis [184,185]. As *EZH2* had been shown to interact with several lncRNAs, such as *HOTAIR* [173] and *PCAT-1* [186], it was unclear which lncRNA was important for *EZH2*-driven prostate cancer progression. Ultimately, knockdown experiments demonstrated that *EZH2-MALAT1* association played a significant role in cancer progression [182], thus representing a new alternative target for treating prostate cancer.

To overcome the low specificity of RIP, crosslinking and immunoprecipitation (**CLIP**) was developed by Ule et al. [187]. CLIP involves the usage of ultraviolet (UV) light to form covalent bonds between RBPs and their direct binding RNAs. An advantage in itself since UV does not crosslink proteins to each other, significantly improving its specificity. In **CLIP-Seq**, after crosslinking, RNA is fragmented, purified and prepared for sequencing [188]. This has led CLIP-Seq to be accepted as a gold standard for identification of endogenous RNA–protein interactions [189].

Since the development of CLIP-Seq, there have been major advancements in CLIP methods that further increase specificity. The first is the development of hybrid CLIP (**hiCLIP**), which enables identification of RNA duplexes bound to RBPs. This is achieved by ligating the two RNA strands with an additional RNA adaptor, following that, the RNA duplexes are immunoprecipitated and sequenced. This method was used to identify mRNA–mRNA and mRNA–lncRNA duplexes bound by Staufen 1 [190]. The second major improvement to CLIP is individual nucleotide CLIP (**iCLIP**), which maps RBP binding sites at nucleotide resolution. A limitation of CLIP is that cDNAs prematurely truncate before the crosslinked nucleotide [191]. However, iCLIP exploits this limitation through the addition of a second adaptor to the 3′ end of cDNA after reverse transcription via circularization [192]. This enables prematurely truncated cDNA at the crosslinked nucleotide to be amplified and therefore improves sensitivity.

#### 3.2.2. RISC Dependent RNA Interactions

The active miRNA research field has led to the development and improvement of a number of methods for establishing which RNAs are being targeted by miRNAs. Thus, establishing whether candidate lncRNAs are involved in RNAi mediated regulation can provide valuable insight into their function. In the first case, lncRNAs targeted by a RISC that is loaded with a complementary miRNA or siRNA may act as decoys or competitive endogenous RNAs (ceRNAs). Circ-lncRNAs may be particularly ideal as ceRNAs due totheir increased stability [38,107]. Colloquially these lncRNAs or circ-lncRNAs are said to “sponge” away interference from targets with other cellular functions, such as mRNAs [193]. Alternatively, lncRNAs may act as the precursors to miRNAs or siRNAs—a further processing step that may be mediated by other RNA binding proteins, such as HuR. These opposing roles can be determined primarily by whether the lncRNA co-occupies the RISC with suspected RNA targets/loads by capturing the ribonucleic or protein part of the complex.

As a first approach, it is possible to identify RNA–RNA interactions on the basis of co-occupation of the RISC complexes isolated via RIP or CLIP based techniques introduced earlier. **Ago2-RIP-Seq** and **Ago-HITS-CLIP** (also called **Ago-CLIP-Seq**) in particular focus on applying RIP-Seq and CLIP-Seq respectively to AGO2, pulling down all miRNAs and possible targets in a single experiment [194,195]. Photoactivatable-Ribonucleoside-Enhanced Crosslinking and Immunoprecipitation (**PAR-CLIP**) is another popular variant of CLIP-Seq that uses photoactivatable nucleoside analogues, such as 4-thiouridine (4SU) to crosslink RISC proteins, such as AGO2 or TNRC6 to the labelled RNAs [121,196]. A particular result of this method is the T to C transitions that occur at the crosslinking sites that can be used to enhance downstream analyses. This technique has been widely cited and implemented including in cancer research [197].

An alternative approach involves utilising a modified ribonucleotide probe to bait and capture any complementary RNAs when they are loaded in the RISC complex. This can be helpful to identify which miRNA/siRNAs are being sponged by a lncRNA acting as ceRNA (competitive endogenous RNA). Additionally, if a lncRNA is suspected of being processed into a miRNA/siRNA, a probe mimicking the lncRNA-derived miRNA/siRNA can be prepared to enable target identification. The first method to apply this concept employed biotinylated miRNA mimic probes to capture their targets in vivo [198]. An in vitro version employing digoxigenin instead has also shown similar performance and is known as the “labelled miRNA pull-down” (**LAMP**) assay system [199]. Many elaborate modifications have been devised to enhance the probes with interesting properties to better capture RNA-RNA duplexes.

A major type of enhancement to the original biotinylated approach has been the inclusion of photoactivatable tags or analogues into the probes (similar to PAR-CLIP). For example, **miR-TRAP** incorporates a psoralen analogue allowing photoactivatable crosslinking to targets, and more stringent purification [200]. The original miR-TRAP method has also recently been paired with RNA-Seq in **PCP-Seq** [201]. PCP-Seq was validated in A549 cancer cell lines.

Other crosslinking technologies incorporated into probes also alter the tag used for isolation procedures. “*Photoclicking*” a process borrowed from bioorthogonal protein chemistry uses tetrazole-ene or dibenzocyclooctyne (DBCO) [202]. **DBCO-tagged** mimic probes in particular have been reported to confer increased miRNA-RISC loading affinity and can be isolated via azide-immobilised magnetic beads [203]. In the **PA-miRNA** method, the biotin tag normally used for isolation is attached via a photo-cleavable linker. It is unclear whether this provides a particular advantage in identifying complementary targets, but the modification is claimed to allow the probe to be used as a photoactivatable source of miRNA [204]. **TargetLink** is a tagless method that utilises a Locked Nucleic Acid (LNA)-based probe for capturing crosslinked miRNA-mRNA complexes and was tested on a human colorectal cancer cell line yielding 12 target genes for miR-21 [205].

Similar approaches can be further enhanced by combining with the technologies used for RNA baiting. Such is the case for **miR-CATCH** which targets a single mRNA (or lncRNA) using a biotinylated DNA probe and crosslinked RISC ribonucleoprotein complexes to detect all miRNAs targetors [206,207]. **miR-CLIP** instead focuses on a single miRNA-like probe containing psoralen and biotin groups to capture the “targetome” after subsequent Ago2 immunoprecipitation and streptavidin purification followed by RNA sequencing [208]. Both techniques have shown promise in cancer research, miR-CATCH has been applied to *MSLN* mRNA, which is overexpressed in Malignant Pleural Mesothelioma and miR-CLIP was validated in Hela cells revealing the lncRNA H19 as a target of miR-106a [209].

Nucleoside analogues, such as diazirine and aryldiazirine, have shown promising results as a means of crosslinking RNA-RNA molecules and post-crosslink tagging has been developed [210,211,212]. Crosslinking chemistry is an active area of research that promises to deliver many more options that may give rise to further variations of RNA pull down methods [213]. Most interestingly, diazirine has even been encoded as an unnatural amino acid, which may open up new interesting possibilities for protein mediated interaction capture [214].

#### 3.2.3. RISC Independent RNA Interactions

Although RISC independent lncRNA–RNA interactions may be less well known, they have been shown to regulate important biological processes, such as somatic tissue differentiation via post-transcriptional mechanisms [215] and cancer cell growth [216]. Interestingly, the extent of base-pairing could determine if the LncRNA–mRNA interactions positively or negatively regulate gene expression. LncRNAs associated with mRNAs through partial base-pairing have been found to promote mRNA decay [217] while more complete base-pairing protects the mRNA from degradation [218].

Many RISC independent RNA–RNA interactions may involve the participation of an RBP other than AGO2. Therefore, a similar RIP or CLIP-based approach targeting other known ribonucleoproteins may allow identification of other proximal RNAs interacting with the lncRNA of interest. Additionally, **CLASH** (cross-linking ligation and sequencing of hybrids), a modified version of iCLIP, is another technique that allows for identification of RNA–RNA interactions by using a tagged “bait” protein [219]. After UV crosslinking of RNA–protein interactions, the bait is pulled down and RNA is recovered and sequenced. When a particular RBP is not targeted **MARIO** (Mapping RNA interactome in vivo) allows EZ-link biotinylation of the protein [220]. A biotinylated RNA-linker is then ligated in a similar fashion to hiCLIP allowing RNA–RNA interactions mediated by other proteins to be captured and sequenced. However, without an RNA binding protein to mediate the interaction, other solutions are required.

Given that lncRNA expression tends to be lower than mRNAs, the levels of endogenous lncRNA must be considered otherwise there might be insufficient material being pulled down. One way to overcome this technical limitation is to perform **in vitro transcribed biotin tagged mimics** of the lncRNA prior to pull down. Not only does this ensure sufficient lncRNA for the pull down but can also improve specificity rather than relying on antisense DNA probe binding. This technique was used in characterising the function of antisense lncRNA of MACC1 in gastric cancer [221]. In that study, bioinformatics predictions suggested that MACC1-AS1 contained a binding site for MACC1 mRNA and the interaction between the two RNAs was later validated via qRT-PCR.

**RIA-seq** (RNA interactome analysis and sequencing) allows for mapping of transcriptome-wide RNA-RNA interactions before selectively probing for your lncRNAs of interest [215]. In brief, cells are fixed with 1% glutaraldehyde before lysis. The RNA are then sonicated to a size range of 100 to 500 nucleotides before addition of antisense DNA probes. The probes are biotinylated and target specific regions of the lncRNA of interest. Thereafter, streptavidin binding captures the beads–biotin-probes–RNA complexes. The RNA is then eluted and qRT–PCR is used to detect enriched transcripts. Alternatively, high-throughput sequencing can be used though sufficient read depth is required to detect interaction. This technique was used to discover a novel mechanism of lncRNA-mRNA interaction in colorectal cancer. The cytoplasmic lncRNA SNHG5 was found to interact with and stabilise their target mRNAs by protecting them from degradation by STAU1. As such, it promotes colorectal cancer cell survival [222]. However, the specificity of RIA-Seq depends largely on the probe design.

Finally, ribonucleoprotein agnostic methods exist to perform transcriptome wide identification of all RNA complexes without specific baits or probes. **PARIS** (Psoralen Analysis of RNA Interactions and Structures), for example, combines psoralen crosslinking with proximity ligation to identify interactions and structural information of all RNAs [223]. In brief, live cells are UV crosslinked and lysed before RNA is extracted and fragmented. PAGE gel electrophoresis is then used to purify RNA where only RNA duplexes are obtained. The RNA duplexes then undergo proximity ligation followed by photo-decrosslinking before the RNA is prepared for sequencing. This technique allows for identification of long-range RNA structures ranging from 200 to over 1000 nt [224]. Apart from detecting just intramolecular interactions and structures, PARIS has been reported to also identify and refine RNA–RNA interactions to near base pair resolution. In addition, unlike other techniques that require specific RNA baits, PARIS allows for identification of native base-pairing interactions through cross-linking of live cells.

Another similar technique is **LIGR-seq** (LIGation of interacting RNA followed by high-throughput sequencing) which uses a psoralen derivative aminomethyltrioxalen (AMT) that intercalates into the RNA for UV crosslinking [225,226]. circRNA ligase is used for proximity ligation of RNA before sequencing. Unlike the PARIS protocol, enrichment of RNA complexes occurs through RNase R digestion of uncrosslinked RNAs. **SPLASH** (Sequencing of Psoralen crosslinked, Ligated, and Selected Hybrids) might be seen as a more robust variation of PARIS and LIGR-seq as it utilises biotin-labelled psoralen for enrichment of crosslinked RNA using streptavidin beads [227].

**RIC-Seq** (RNA in situ conformation sequencing) has also recently entered this arena of whole RNA-interactome and secondary structure mapping [228]. Similar to SPLASH, in situ proximity ligation of RNA complexes is applied and biotin enables pulldown of crosslinked RNA. However, it substitutes psoralen with pCp thereby labelling the 3′ end of RNA [229] instead of staggered pyrimidines on opposite strands [230]. This step would seem to give RIC-Seq an edge over psoralen-based techniques as it appears to more effectively enrich RNA complexes allowing for detection of lowly expressed RNA [228].

Fundamentally, all these RNA–RNA interactome methods apply proximity ligation with key differences at the RNA-complex isolation steps. Interpreting the results from these largescale experiments is challenging especially considering the current bioinformatic tools are still somewhat underdeveloped. Effectively, analyses borrow tools traditionally used for HiC. However, they have all shown promise in being able to identify lncRNA interactions. For example, *MALAT1* was found to interact with *NEAT1* through analysis of RIC-Seq data [228]. Nevertheless, the aforementioned methods (summarised in Table 4) are providing valuable results that are being compiled into databases such as RISE and can be re-analysed using updated analytical tools when available to continue improving our understanding of RNA–RNA interactomes and structure. Such results can ultimately be followed by more precise experiments to reveal the regulatory effects of the lncRNA interaction.

### 3.3. Other Approaches and Biochemical Assays

Whether potential candidate lncRNA and interacting partners have been predicted using bioinformatic tools or via high-throughput sequencing techniques, it is also possible to apply other low- or medium-throughput technologies to characterise or further validate possible interactions.

#### 3.3.1. Protein Interaction Assays

**Microarrays** provide an alternative method to Next-Generation Sequencing (NGS) for lncRNA-protein interaction studies. This relatively inexpensive method is able to provide information in a couple of hours. However, it has limitations in identifying novel RNA targets. It could be used to quantify and identify either annotated RNA targets or RBPs in ribonucleoprotein (RNP) complexes. RNP immunoprecipitation–microarray (RIP-Chip) has been applied successfully in detecting several QKI-5-binding lncRNAs, especially *lnc10* that regulates the apoptosis of germ cells during their development [231]. In this method, the crosslinking reaction may be omitted during RIP as cell extracts will be used to identify RBPs. However, the crosslinking step can give results with high backgrounds and introduce sequence biases. Briefly, cell extracts are used for immunoprecipitation against the protein of interest and then washed extensively, following which the RNP is eluted and dissociated into RNA and protein [232]. Besides this, protein microarrays have also been widely used to detect RBPs that interact with certain lncRNA. Here, lncRNAs are transcribed in vitro and labelled with Cy5 dye, then labelled lncRNAs are incubated with a protein microarray [233]. Protein microarrays have been able to detect the interaction between *TINCR* RNA and STAU1 protein [215]. Aberrant expression of *TINCR* RNA is implicated in the progression of many cancers. *TINCR* RNA overexpression in epithelial ovarian cancer has been reported to correlate with tumour size, metastasis and survival rates in the patients. By silencing *TINCR*, FGF2 expression is downregulated and leads to the inhibition of epithelial ovarian cancer progression [234].

**Dot-blot assay** is widely used to study lncRNA–protein interaction and is especially useful in mapping the protein binding region in lncRNA. In this assay, lncRNAs of interest are biotinylated and transcribed in vitro, followed by in vitro RNA-protein binding via incubation of the biotinylated lncRNA with recombinant protein. The bound lncRNA is partially digested by RNase to allow only a small fragment attached to the protein. The lncRNA–protein complexes are subjected to proteinase K treatment to dissociate the complexes. Subsequently, the lncRNA is purified and hybridized to nylon or PVDF membranes spotted with 54–60 mer antisense DNA oligonucleotides tiled along the lncRNA of interest [235,236]. The hybridized membrane is washed and visualized by the detection of streptavidin-HRP signals. This assay has successfully identified the motif sequence of *BCAR4* bound by SNIP1 and PNUTS, which is located at positions 235–288 and 991–1044 in *BCAR4* [236]. In a tumour microenvironment study, positions 355–414 and 1298–1353 of lncRNA *CamK-A* are bound and protected by PNCK and IκBα, which is important in tumour progression [237].

**Mass spectrometry (MS)** is commonly used to characterize various proteins that are associated with lncRNAs, following pull-down of the lncRNA of interest. It can identify and quantify molecules in complex mixtures based on their mass and charge. However, the quantification accuracy may not be correct due to the difference in mass spectrometric responses. To overcome this issue, stable-isotope labelling has been applied before proceeding with MS. The stable-isotope labelling by amino acids in cell culture (SILAC) has been shown to simplify the quantification and remove false-positive results [238]. This labelling is performed by simply growing two cell populations in two different mediums containing either light or heavy amino acids. Then, the cells are mixed, and proteins extracted for MS analysis. This method has been used to identify several proteins that are specifically enriched and found to interact with *Xist* to mediate transcriptional silencing [239]. Moreover, aberrant expression of *Xist* is associated with tumour progression and metastasis in multiple cancers. Knockdown of *Xist* in colorectal cancer has been proven to inhibit cell proliferation, invasion, and epithelial-mesenchymal transition (EMT) [240]. Larger tumour size and advanced stage of tumour are significantly correlated with high expression of *Xist*. Hence, *Xist* expression is used to predict the prognosis and survival of colorectal cancer patients [241].

#### 3.3.2. RNA Interaction Assays

**Co-sedimentation assays** can be used whereby RNA is extracted from cells and fractionated using sucrose or glycerol gradients. The RNAs found in the different fractions are examined by Northern blot. RNAs found in the same gradient fractions are thought to interact with each other [242,243] though it does not directly demonstrate interaction. A more robust experiment would be the electrophoretic mobility shift assay (EMSA), which involves studying interaction of RNA fragments by observing rate of migration of the samples during gel electrophoresis [244]. If the lncRNA–mRNA interacts, the complex would have a larger molecular mass compared to separate strands of RNA. Therefore, the complex would migrate slower on the gel compared to non-paired RNAs. Samples can be extracted from cells or synthesised in vitro. Synthesising of RNA fragments could potentially demonstrate interaction between specific regions of the lncRNA-mRNA complexes. However, these techniques can only screen for a given set of molecules.

**Ribonuclease protection assays (RPA)** can also be used to detect these sense-antisense RNA duplexes. RPA involves isolation of total RNA followed by RNase and DNase digestion [245,246,247]. Duplexed RNA should be protected from digestion and will be detected by PCR and gel electrophoresis or qRT-PCR. This technique was used to demonstrate interaction between PDCD4-AS1 and PDCD4 mRNA in breast cancer [248].

**Microarrays** as mentioned earlier have been used to identify the alternative splicing (AS) events regulated by MALAT1 for example. PolyA+ RNA isolated from MALAT1-antisense oligo treated and control HeLa cells were isolated and prepared into labelled cDNA. This was hybridized to a custom AS microarray. The GenASAP algorithm was then used to estimate the percent exon inclusion. Semiquantitative RT-PCR using primers specific for exons flanking the AS events was performed to validate the microarray predictions. This assay revealed that MALAT1 depleted cells have changes in AS of B-MYB and MGEA6 pre-mRNAs [177].

## 4. Closing Remarks

Throughout this review, we have introduced both computational resources and experimental methods to perform primary and secondary characterisations of lncRNAs to ascertain their potential roles in post-transcriptional regulation and cancer. Primary characterisation establishes basic ground truths relating expression, localisation and relative importance in a model system of choice under normal or perturbed conditions. Secondary characterisation particularly focuses on identifying the interacting RNAs and RBPs that the lncRNAs may influence and importantly deregulate in cancer states due to aberrant expression or non-coding mutations affecting their binding. The RIP and CLIP technologies in particular have been well adapted to next-generation sequencing allowing these newer methods to become reference options for performing high-throughput screening of RNA/Protein or RNA/RNA interactions. Furthermore, a host of improvements to the cross-linking biochemistry have been incorporated and enabled progressive advances in specificity and reliability. It will be important to continue generating these types of experiments and complementing the growing databases dedicated to cataloguing and integrating this information with other valuable sources as presented in Section 2.1, which are already useful starting points for orienting experimental strategies for more novel lncRNAs. The appearance of disease and cancer specialised lncRNA databases and metadatabases will prove highly valuable in integrating the diverse interaction sources and placing them in relevant cancer contexts to identify interesting regulatory patterns or correlations in diseased states.

Some of the findings concerning lncRNA functions in cancer may have direct applications for therapies involving Antisense oligonucleotides (ASOs) for example. ASOs are DNA:RNA chimeras that direct RNase H to degrade target RNAs [249] such as target lncRNAs associated with cancer. During preclinical trials, they were able to target *MALAT1* in vivo, resulting in reduced metastasis [250]. Additionally, ASOs can operate through other mechanisms, such as steric blocking of TF binding and modulating splicing [251]. Unfortunately, RNA molecules like lncRNAs are able to form multiple conformations given their intrinsic flexibility [252]. This makes predicting their structure a challenge which could impede the success of targeted regulation of their expression.

As non-coding RNA continues to take on importance in influencing fundamental processes such as post-transcriptional regulation it will be interesting to integrate this knowledge with findings relating to other novel post-transcriptional regulatory mechanisms such as RNA modifications. This field has also benefited from the adoption of CLIP technologies to perform epitranscriptomic studies on some of the hundreds of modifications that are likely to affect RNA stability, structure, localisation and interactions—lncRNAs included [253,254,255]. All of these CLIP-based sequencing methods will continue evolving with the maturation of third-generation sequencing, which is already enabling native RNA sequencing including RNA modification detection and structural footprinting [256,257]. In the near future, it may well be possible to capture RNA interactomes, methylomes and structuromes in single experiments to reveal a more complete landscape of the post-transcriptional regulatory mechanisms susceptible to exploitation by cancers.

## Figures and Tables

**Figure 1 ncrna-07-00019-f001:**
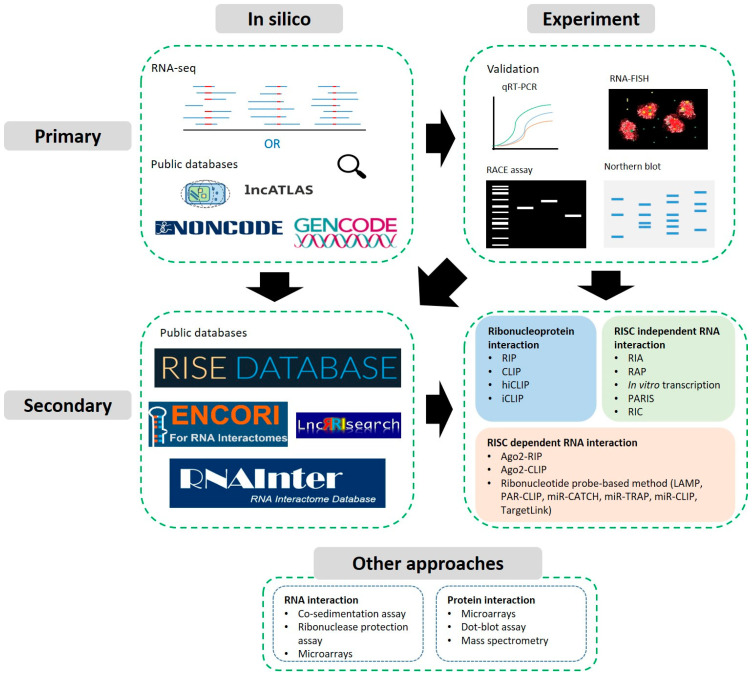
Workflow(s) for the detection and functional characterisation of a lncRNA of interest and its interacting partners. Primary approaches focus on identifying and assigning basic properties based on existing knowledge, predictions or biochemical experiments to validate expression or localisation for example. Secondary approaches focus on the identification of interactions with RNAs or RBPs utilising sequencing-based techniques, see Section 3.2. Further approaches may also be useful to validate high throughput or predictive results.

**Table 1 ncrna-07-00019-t001:** Databases for identifying lncRNAs and their basic properties or associations. Overview of active databases cataloguing various properties (sequence conservation, mutation, expression, localisation) or associations attributed to lncRNA genes or transcripts. A link to the hosting website is provided followed by the latest known version as well as the most recent publication describing the database.

Database/Version/Ref.	Link	Conservation	Mutations	Expression	Localisation	Associations
LNCiPedia v5 (2019) [89]	https://lncipedia.org/ (accessed on 8 March 2021)	*H. sapiens, D. melanogaster, D. rerio, M. musculus, P. troglodytes*	NA	NA	NA	Relevant references
lncATLAS (2017) [90]	https://lncatlas.crg.eu/ (accessed on 8 March 2021)	NA	NA	**GENCODE**	**GENCODE**	NA
NONCODE v6 (2020) [88]	http://www.noncode.org/ (accessed on 8 March 2021)	*H. sapiens, M. musculus* and 15 more	**dbSNP**	**Human Body Map; NCBI GEO**	NA	**Gene Ontology**
lncWiki/Book (2019) [91,92]	https://bigd.big.ac.cn/lncrnawiki/index.php/Main_Page (accessed on 8 March 2021) https://bigd.big.ac.cn/lncbook/index (accessed on 8 March 2021)	NA	**ClinVar; COSMIC**	**HPA; GTEx;** Methylation	NA	**Gene Ontology; MeSH Ontology;** miRNA Interaction Prediction;
Lnc2Cancer v3 (2020) [93]	http://bio-bigdata.hrbmu.edu.cn/lnc2cancer/ (accessed on 8 March 2021)	NA	NA	Literature Mining	lncATLAS	Expression Correlation; Survival; TF Motif; *lncBook*
LncRNADisease v2 (2019) [94]	http://www.rnanut.net/lncrnadisease/ (accessed on 8 March 2021)	NA	NA	NA	NA	**Disease Ontology; MeSH Ontology;** Predictive Associations
LncMAP v2 (2018) [95]	http://bio-bigdata.hrbmu.edu.cn/LncMAP/ (accessed on 8 March 2021)	NA	NA	NA	NA	Associations with: TF, Genes, Drugs, Survival
TANRIC v2 (2019) [96]	https://www.tanric.org (accessed on 8 March 2021)	NA	**TCGA** Somatic Mutations	**TCGA**	NA	**TCGA/CCLE** Correlations: Expression, Stage; Survival
MNDR v3.1 (2020) [97]	https://www.rna-society.org/mndr/ (accessed on 8 March 2021)	NA	NA	Mammalian	NA	Evidenced disease associations and Predictor
lncRNASNP v2 (2018) [98]	http://bioinfo.life.hust.edu.cn/lncRNASNP/#!/ (accessed on 8 March 2021)	NA	**TCGA** and **COSMIC** SNVs	NA	NA	miRNA binding & SNP effects; GWAS LD; Mutation effects
lncRNAMAP (2014) [99]	https://lncrnamap.mbc.nctu.edu.tw (accessed on 8 March 2021)	NA	NA	**NCBI GEO**	NA	miRNA and endo-siRNA predictors
LncTarD (2020) [100]	http://bio-bigdata.hrbmu.edu.cn/LncTarD/ (accessed on 8 March 2021)	NA	NA	NA	NA	Disease-related Target Prediction
EVLncRNAs (2017) [101]	http://biophy.dzu.edu.cn/EVLncRNAs/ (accessed on 8 March 2021)	NA	NA	NA	NA	Manually curated disease association
LncSPA (2020) [102]	http://bio-bigdata.hrbmu.edu.cn/LncSpA/ (accessed on 8 March 2021)	NA	NA	**GTEx, HPA, HBM2, FANTOM, TCGA, TARGET**	NA	Expression in diseased tissues

**Table 2 ncrna-07-00019-t002:** Databases for identifying circRNAs and their basic properties or associations. The table summarises for each database what species the data are based on, as well as data sources, integrations or predictions of circRNA genes or transcripts. A link to the hosting website is provided followed by the latest known version as well as the most recent publication describing the database. IRES and MRE correspond to Internal Ribosome Entry Sites and miRNA Response Elements. See Section 3.2 for more information on CLIP and PAR-CLIP techniques.

Database/Version/Ref.	Link	Species	Data Sources	Integrations	Predictions
CircAtlas (2020) [103]	http://159.226.67.237:8080/new/index.php (accessed on 8 March 2021)	*H. sapiens, M. mulatta, M. musculus, R norvegicus, S. scrofa and G gallus*	1070 RNA-seq samples across 6 species	Integrates **circR2Disease** and **circRNADIsease** for disease associations	Co-expression network; Functional inference from **GO/KEGG;** RBP and miRNA binding
circRNAdb (2016) [104]	http://reprod.njmu.edu.cn/cgi-bin/circrnadb/circRNADb.php (accessed on 8 March 2021)	*H. sapiens*	Literature and RNA-seq dataset	**UniProt**	Protein domains, post-translational modifications, half-lifes
CircFunBase (2019) [105]	http://bis.zju.edu.cn/CircFunBase/ (accessed on 8 March 2021)	*H. sapiens, M. musculus + 13 more.*	Literature search	**CircInteractome** (CLIP data), **miRBase**	miRNA-circRNA interactions
circBase (2017) [106]	http://www.circbase.org/ (accessed on 8 March 2021)	*H. sapiens, C. elegans, D. melanogaster, M. musculus, L. chalumnae, L. menadoensis*	Various publications [18,107,108,109,110,111]	**doRiNA**	NA
Circbank (2019) [112]	http://www.circbank.cn/ (accessed on 8 March 2021)	*M. musculus, R. norvegicus, D. melanogaster*	**circBase, miRBase**	m6A literature, COSMIC somatic mutations	IRES, circRNA-miRNA prediction
CIRCpedia v2 (2018) [113]	https://www.picb.ac.cn/rnomics/circpedia/ (accessed on 8 March 2021)	*H. sapiens, M. musculus, R. norvegicus, D. rerio, D. melanogaster, C*	180 RNA-seq samples across 6 species	NA	Putative circRNAs
CircRNADisease (2018) [114]	http://cgga.org.cn:9091/circRNADisease/ (accessed on 8 March 2021)	*H. sapiens*	Manual curation of 800 publications	NA	Association to diseases
CircR2Disease (2018) [115]	http://bioinfo.snnu.edu.cn/CircR2Disease/ (accessed on 8 March 2021)	*H. sapiens*	Manual curation of literature	NA	Association to diseases
TSCD (2017) [116]	http://gb.whu.edu.cn/tscd/ (accessed on 8 March 2021)	*H. sapiens, M. musculus*	**ENCODE** + NCBI **GEO** RNA-seq	**Starbase, Gene Ontology**	MRE, Protein binding sites
circad (2020) [117]	http://clingen.igib.res.in/circad/ (accessed on 8 March 2021)	*H. sapiens, M. musculus, R. rattus*	Manual curation of literature	NA	Asssociation to diseases
circVAR (2020) [118]	http://soft.bioinfo-minzhao.org/circvar/ (accessed on 8 March 2021)	*H. sapiens*	**circBase, circNet, circRNAdb**	**1000 Genomes, ClinVAR, GWASCatalog, ClinVAR, COSMIC**	Association to diseases/cancer
CSCD (2018) [119]	http://gb.whu.edu.cn/cscd/ (accessed on 8 March 2021)	*H. sapiens*	228 RNA-seq samples from **ENCODE**	**Starbase**	Cancer Association, MRE, RBP, ORFs
Circ2Traits (2013) [120]	http://gyanxet-beta.com/circdb/ (accessed on 8 March 2021)	*H. sapiens*	RNA-seq [107]	**Starbase, TargetScan, miRCode, dbSNP, GWAS catalog,** PAR-CLIP Data [121]	miRNA interactions
Circ2Disease (2018) [122]	http://bioinformatics.zju.edu.cn/Circ2Disease/index.html (accessed on 8 March 2021)	*H. sapiens*	Manual curation of literature	**HMDD, OncomiRDB, miRTarBase, dbDEMC, miRecords**	miRNA interactions
CircInteractome (2016) [123]	https://circinteractome.nia.nih.gov/ (accessed on 8 March 2021)	*H. sapiens*	**circBase**	**Starbase, miRBase**	IRES, RBP and miRNA binding sites

**Table 4 ncrna-07-00019-t004:** Summary of sequencing approaches for facilitating the characterization of lncRNAs.

Method	Specifications	Limitations	Requirements (Time/Special Resources)
**RIP/RIP-seq** [182] (tagged/endogenous RBP mediated RNA co-occupancy)	Characterization of native RNA-protein complexes without crosslinking; antibody enrichment	Low specificity; dependent on antibody availability	3–4 d/IP compatible antibody; Autoradiograph facilities
**CLIP/CLIP-seq** [187] (tagged/endogenous RBP mediated RNA co-occupancy)	RNA-protein interaction sites via RNA-Protein UV crosslinking; antibody enrichment	5′ and 3′ sites of RNA tags affected by cleavage and ligation biases; dependent on antibody availability	5–8 d/IP compatible antibody; UV Crosslinker; Autoradiograph facilities
**hiCLIP** [190] (tagged/endogenous RBP mediated RNA co-occupancy and RNA-duplexes)	RNA-protein interaction sites and RNA duplexes via UV crosslinking; antibody enrichment	May only capture highly expressed RNA species; dependent on antibody availability	5 d/IP compatible antibody; UV Crosslinker; Autoradiograph facilities
**iCLIP** [192] (tagged/endogenous RBP mediated RNA co-occupancy)	RNA-protein interaction sites at nucleotide resolution via UV crosslinking; antibody enrichment	miRNA-target interaction strength; dependent on antibody availability	5 d/IP compatible antibody; UV Crosslinker; Autoradiograph facilities
**PAR-CLIP** [121] (tagged/endogenous RBP mediated RNA co-occupancy)	RNA-protein interaction sites at nucleotide resolution; enhanced UV cross-linking and analysis choices; antibody enrichment	cultured cells only; 4-SU can induce cellular stress; dependent on antibody availability	5 d/IP compatible antibody; UV Crosslinker; Autoradiograph facilities
**Biotin-mimics/LAMP** [199] (tagged miRNA mimic probing RNA targets)	One miRNA to many RNA interactions; Biotin enrichment	Delivered by transfection to cultured cells; Requires known miRNA sequence	2 d/Streptavidin magnetic beads
**miR-TRAP/PCP-seq** [200,201] (tagged miRNA mimic probing RNA targets)	One miRNA to many RNA interactions at nucleotide resolution; UVA crosslinking; Poly-A enrichment	Delivered by transfection to cultured cells; Requires known miRNA sequence	2–3 d/UV Crosslinker
**DBCO-tagged mimics** [203] (tagged miRNA mimic probing RNA targets)	One miRNA to many RNA interactions; increased loading affinity; Click enrichment	Requires known miRNA sequence	3 d/Azide-immobilized magnetic Beads
**PA-miRNA** [204] (tagged miRNA mimic probing RNA targets)	One miRNA to many RNA interactions; Photocleavable linker; Biotin enrichment	Delivered by transfection to cultured cells; Requires known miRNA sequence; linker is not easily acquired	5 d/Solid phase synthesis; HPLC; Mass spectrometry; UV Crosslinker; Streptavidin magnetic beads
**TargetLink** [205] (tagged miRNA mimic probing RNA targets)	One or more miRNAs to many RNA targets; LNA+Biotin enrichment	Requires KO control; Requires known miRNA sequence	6 d/UV Crosslinker; HPLC; Streptavidin magnetic beads
**miR-CATCH** [206] (tagged RNA mimic probing miRNA targetors)	One RNA to many miRNA interactions; RNA-RISC crosslinking by formaldehyde; Biotin enrichment	Delivered by transfection to cultured cells; Requires known RNA sequence	3–4 d/Dynamag-2; FastPrep-24; Hybridization Oven; Streptavidin magnetic beads
**miR-CLIP** [208] (tagged miRNA mimic probing RNA targets)	One miRNA to many RNA interactions; RNA-RNA crosslinking by psoralen; Biotin enrichment	Delivered by transfection to cultured cells; Requires known miRNA sequence; Probe needs testing	3–4 d/HPLC; UV Crosslinker; Streptavidin magnetic beads
**CLASH** [219] (tagged RBP mediated RNA-Protein/duplex capture)	RNA-protein interaction sites and RNA duplexes via UV crosslinking; IgG+Ni-NTA enrichment	Delivered by transfection to cultured cells; Tagged protein expression design may be challenging	4–5 d/UV Crosslinker; Autoradiography facilities
**MARIO** [220] (endogenous RBP mediated RNA-duplex capture)	Global RNA-RNA interactions mediated by RBPs; RNA-Protein UV crosslinking; 2-step biotin enrichment; proximity ligation	Limited to RBP mediated interactions	5 d/UV Crosslinker; Streptavidin magnetic beads
**RIA-seq** [215] (endogenous RNA-duplex capture)	One RNA to all RNA interactions; glutaraldehyde crosslinking; biotin enrichment	Limited to RBP mediated interactions; probe preparation may be challenging	5 d/Streptavidin magnetic beads
**PARIS** [223] (endogenous RNA-duplex capture)	All to all RNA interactions; psoralen crosslinking of RNAs; 2D enrichment of crosslinked duplexes; proximity ligation	Possible AMT side effects; 2D gel setup may be challenging	5 d/UV Crosslinker; SequaGel UreaGel System
**LIGR-seq** [225] (endogenous RNA-duplex capture)	All to all RNA interactions; psoralen crosslinking of RNAs; RNAseR enrichment of crosslinked duplexes; proximity ligation	Possible AMT side effects	4 d/UV Crosslinker; RNAseR
**SPLASH** [227] (endogenous RNA-duplex capture)	All to all RNA interactions; psoralen crosslinking of RNAs; biotin enrichment of crosslinked duplexes; proximity ligation	Possible AMT side effects	4 d/UV Crosslinker; Streptavidin magnetic beads
**RIC-seq** [228] (endogenous RBP mediated RNA-duplex capture)	Global RNA-RNA interactions mediated by RBPs; RNA-Protein formaldehyde crosslinking; biotin enrichment; in situ proximity ligation	Limited to RBP mediated interactions; cell permeabilization may need optimizing	5 d/Streptavidin magnetic beads

## Data Availability

No new data were created or analyzed in this study. Data sharing is not applicable to this article.

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
