# Peer review of "Approaches to Identify and Characterise the Post-Transcriptional Roles of lncRNAs in Cancer"

_ncrna, 2021, doi:10.3390/ncrna7010019_

Round 1

Reviewer 1 Report

The paper is well written and fulfills its role of guide researchers that wants to begin to study ncRNAs. 

One point that could be improved in the paper is a figure or table summarizing all the cited methods indicating their applications, pros and cons, besides their costs (high/low) and the type of equipment needed. With this, the text could be reduced a lot, making the paper more attractive.

Author Response

Please see the attachment, many thanks.

Reviewer 2 Report

Carter et al. provided a systematic review of current computational resources and experimental approaches used to identify and characterize the lncRNA roles in the post-transcriptional regulatory layer. In general, this article not only combined, organized, and distilled knowledge from lots of existing works in a concise way, but offered critical insights to help advance our understandings of how lncRNAs are involved in tumorigenesis. Only a few recommendations given below.

First, lncRNAs are composed of both linear and circular forms. The authors completely left circRNAs behind their review.

Second, TCGA Pan-Cancer Analysis and other studies had provided computational prediction methods to infer lncRNAs as ceRNAs. Please survey and include this type of works into your article.

Third, lncRNAs can compete with other mRNAs/lncRNAs for RBP binding as well. Please include this into your ceRNA section.

Fourth, the majority of linear lncRNAs are transcriptional regulators, but circRNAs are often exerting their regulatory functions post-transcriptionally. The authors should clarify this point and provide relevant references.

Lastly, please summarize the advantages and disadvantages of existing experimental approaches in a table to help your readers choose the right technology for their research projects.

Author Response

Please see the attachment, many thanks.

Reviewer 3 Report

In this review manuscript, the authors summarizing the current field of lncRNAs in cancers by specifically focusing on tools that are available to analyze cancer lncRNAs. This review manuscript provides a nice overview of the current techniques available in the field to characterize lncRNAs. Although many different techniques, databases, and tools are named in this manuscript, the details of each technique/database/tool are missing. More specific comments are listed below:

Major points:

[1] Lines 27-28: "This fundamental process relentlessly transcribes as much as 60% of the human genome" The authors should check the latest understanding of how much of human genome is transcribed as RNA, which is as high as 90%.

[2] 2.1. Predictions and databases. The authors simply list several NGS techniques (e.g., GRO-Seq), but they never explain such techniques in details. Also, there are no references to these techniques. To make it easier for the readers, each technique should be explain briefly by citing the appropriate references. The same applies to the other parts of the manuscript mentioning different techniques and databases. Although the authors describe some of these techniques in details in the later part of the manuscript, it is still advisable to mention clearly in the text that each of these techniques will be described in detailed in the following sections of this manuscript.

[3] Table 1. The authors should be more specific by naming the organisms rather than simply stating "Major model systems".

Minor points:

(1) All lncRNA gene symbols should be in Italic.

(2) Table 1. The URL of each database should be provided.

Author Response

Please see the attachment, many thanks.

Round 2

Reviewer 2 Report

The authors had fully resolved all issues I pointed out previously. I am satisfied with this revised manuscript.